# RESIDUAL NON-LOCAL ATTENTION NETWORKS FOR IMAGE RESTORATION

**Yulun Zhang[1], Kunpeng Li[1], Kai Li[1], Bineng Zhong[2] & Yun Fu[1,3]**
[1]Department of ECE, Northeastern University, Boston, MA 02115, USA
[2]School of Computer Science and Technology, Huaqiao University, Xiamen 362100, China
[3]College of CIS, Northeastern University, Boston, MA 02115, USA
{yulun100,kinpeng.li.1994,li.kai.gml}@gmail.com,
bnzhong@hqu.edu.cn,yunfu@ece.neu.edu

## ABSTRACT

In this paper, we propose a residual non-local attention network for high-quality image restoration. Without considering the uneven distribution of information in the corrupted images, previous methods are restricted by local convolutional operation and equal treatment of spatial- and channel-wise features. To address this issue, we design local and non-local attention blocks to extract features that capture the long-range dependencies between pixels and pay more attention to the challenging parts. Specifically, we design trunk branch and (non-)local mask branch in each (non-)local attention block. The trunk branch is used to extract hierarchical features. Local and non-local mask branches aim to adaptively rescale these hierarchical features with mixed attentions. The local mask branch concentrates on more local structures with convolutional operations, while non-local attention considers more about long-range dependencies in the whole feature map. Furthermore, we propose residual local and non-local attention learning to train the very deep network, which further enhance the representation ability of the network. Our proposed method can be generalized for various image restoration applications, such as image denoising, demosaicing, compression artifacts reduction, and super-resolution. Experiments demonstrate that our method obtains comparable or better results compared with recently leading methods quantitatively and visually.

## 1 INTRODUCTION

Image restoration aims to recover high-quality (HQ) images from their corrupted low-quality (LQ) observations and plays a fundamental role in various high-level vision tasks. It is a typical ill-posed problem due to the irreversible nature of the image degradation process. Some most widely studied image restoration tasks include image denoising, demosaicing, and compression artifacts reduction. By distinctively modelling the restoration process from LQ observations to HQ objectives, i.e., without assumption for a specific restoration task when modelling, these tasks can be uniformly addressed in the same framework. Recently, deep convolutional neural network (CNN) has shown extraordinary capability of modelling various vision problems, ranging from low-level (e.g., image denoising (Zhang et al., 2017a), compression artifacts reduction (Dong et al., 2015), and image super-resolution (Kim et al., 2016; Lai et al., 2017; Tai et al., 2017; Lim et al., 2017; Zhang et al., 2018a; Haris et al., 2018; Wang et al., 2018c; Zhang et al., 2018b; Wang et al., 2018b)) to high-level (e.g., image recognition (He et al., 2016)) vision applications.

Stacked denoising auto-encoder (Vincent et al., 2008) is one of the best well-known CNN based model for image restoration. Dong et al. proposed SRCNN (Dong et al., 2014) for image super-resolution and ARCNN (Dong et al., 2015) for image compression artifacts reduction. Both SRCNN and ARCNN achieved superior performance against previous works. By introducing residual learning to ease the training difficulty for deeper network, Zhang et al. proposed DnCNN (Zhang et al., 2017a) for image denoising and compression artifacts reduction. The denoiser prior was lately introduced in IRCNN (Zhang et al., 2017b) for fast image restoration. Mao et al. proposed a very deep fully convolutional encoding-decoding framework with symmetric skip connections for image restoration (Mao et al., 2016). Tai et al. later proposed a very deep end-to-end persistent memory network (MemNet) for image restoration (Tai et al., 2017) and achieved promising results. These CNN based methods have demonstrated the great ability of CNN for image restoration tasks.

However, there are mainly three issues in the existing CNN based methods above. **First**, the receptive field size of these networks is relatively small. Most of them extract features in a local way with convolutional operation, which fails to capture the long-range dependencies between pixels in the whole image. A larger receptive field size allows to make better use of training inputs and more context information. This would be very helpful to capture the latent degradation model of LQ images, especially when the images suffer from heavy corruptions. **Second**, distinctive ability of these networks is also limited. Let's take image denoising as an example. For a noisy image, the noise may appear in both the plain and textural regions. Noise removal would be easier in the plain area than that in the textural one. It is desired to make the denoising model focus on textual area more. However, most previous denoising methods neglect to consider different contents in the noisy input and treat them equally. This would result in over-smoothed outputs and some textural details would also fail to be recovered. **Third**, all channel-wise features are treated equally in those networks. This naive treatment lacks flexibility in dealing with different types of information (e.g., low- and high-frequency information). For a set of features, some contain more information related to HQ image and the others may contain more information related to corruptions. The interdependencies among channels should be considered for more accurate image restoration.

To address the above issues, we propose the very deep residual non-local attention networks (RNAN) for high-quality image restoration. We design residual local and non-local attention blocks as the basic building modules for the very deep network. Each attention block consists of trunk and mask branches. We introduce residual block (He et al., 2016; Lim et al., 2017) for trunk branch and extract hierarchical features. For mask branch, we conduct feature downscaling and upscaling with large-stride convolution and deconvolution to enlarge receptive field size. Furthermore, we incorporate non-local block in the mask branch to obtain residual non-local mixed attention. We apply RNAN for various restoration tasks, including image denoising, demosaicing, and compression artifacts reduction. Extensive experiments show that our proposed RNAN achieves state-of-the-art results compared with other recent leading methods in all tasks. To the best of our knowledge, this is the first time to consider residual non-local attention for image restoration problems.

The main contributions of this work are three-fold:

- We propose the very deep residual non-local networks for high-quality image restoration. The powerful networks are based on our proposed residual local and non-local attention blocks, which consist of trunk and mask branches. The network obtains non-local mixed attention with non-local block in the mask branch. Such attention mechanis helps to learn local and non-local information from the hierarchical features.

- We propose residual non-local attention learning to train very deep networks by preserving more low-level features, being more suitable for image restoration. Using non-local low-level and high-level attention from the very deep network, we can pursue better network representational ability and finally obtain high-quality image restoration results.

- We demonstrate with extensive experiments that our RNAN is powerful for various image restoration tasks. RNAN achieves superior results over leading methods for image denoising, demosaicing, compression artifacts reduction, and super-resolution. In addition, RNAN achieves superior performance with moderate model size and performs very fast.

## 2  RELATED WORK

**Non-local prior.** As a classical filtering algorithm, non-local means (Buades et al., 2005) is computed as weighted mean of all pixels of an image. Such operation allows distant pixels to contribute to the response of a position at a time. It was lately introduced in BM3D (Dabov et al., 2007b) for image denoising. Recently, Wang et al. (2018a) proposed non-local neural network by incorporating non-local operations in deep neural network for video classification. We can see that those methods mainly introduce non-local information in the trunk pipeline. Liu et al. (2018) proposed non-local recurrent network for image restoration. However, in this paper, we mainly focus on learning non-local attention to better guide feature extraction in trunk branch.

**Attention mechanisms.** Generally, attention can be viewed as a guidance to bias the allocation of available processing resources towards the most informative components of an input (Hu et al.,

2017). Recently, tentative works have been proposed to apply attention into deep neural networks (Wang et al., 2017; Hu et al., 2017). It's usually combined with a gating function (e.g., sigmoid) to rescale the feature maps. Wang et al. (2017) proposed residual attention network for image classification with a trunk-and-mask attention mechanism. Hu et al. (2017) proposed squeeze-and-excitation (SE) block to model channel-wise relationships to obtain significant performance improvement for image classification. In all, these works mainly aim to guide the network pay more attention to the regions of interested. However, few works have been proposed to investigate the effect of attention for image restoration tasks. Here, we want to enhances the network with distinguished power for noise and image content.

**Image restoration architectures.** Stacked denoising auto-encoder (Vincent et al., 2008) is one of the most well-known CNN-based image restoration method. (Dong et al., 2015) proposed AR-CNN for image compression artifact reduction with several stacked convolutional layers. With the help of residual learning and batch normalization (Ioffe & Szegedy, 2015), Zhang et al. proposed DnCNN (Zhang et al., 2017a) for accurate image restoration and denoiser priors for image restoration in IRCNN (Zhang et al., 2017b). Recently, great progresses have been made in image restoration community, where Timofte et al. (Timofte et al., 2017), Ancuti et al. (Ancuti et al., 2018), and Blau et al. (Blau et al., 2018) lead the main competitions recently and achieved new research status and records. For example, Wang et al. (Wang et al., 2018c) proposed a fully progressive image SR approach. However, most methods are plain networks and neglect to use non-local information.

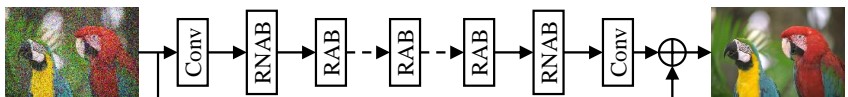

Figure 1: The framework of our proposed residual non-local attention network for image restoration. 'Conv', 'RNAB', and 'RAB' denote convolutional layer, residual non-local attention block, and residual local attention block respectively. Here, we take image denoising as a task of interest.

# 3 RESIDUAL NON-LOCAL ATTENTION NETWORK FOR IMAGE RESTORATION

## 3.1 FRAMEWORK

The framework of our proposed residual non-local attention network (RNAN) is shown in Figure 1. Let's denote $I_L$ and $I_H$ as the low-quality (e.g., noisy, blurred, or compressed images) and high-quality images. The reconstructed image $I_R$ can be obtained by

$$I_R = H_{RNAN}(I_L),\qquad(1)$$

where $H_{RNAN}$ denotes the function of our proposed RNAN. With the usage of global residual learning in pixel space, the main part of our network can concentrate on learning the degradation components (e.g., noisy, blurring, or compressed artifacts).

The first and last convolutional layers are shallow feature extractor and reconstruction layer respectively. We propose residual local and non-local attention blocks to extract hierarchical attention-aware features. In addition to making the main network learn degradation components, we further concentrate on more challenging areas by using local and non-local attention. We only incorporate residual non-local attention block in low-level and high-level feature space. This is mainly because a few non-local modules can well offer non-local ability to the network for image restoration.

Then RNAN is optimized with loss function. Several loss functions have been investigated, such as $L_2$ (Mao et al., 2016; Zhang et al., 2017a; Tai et al., 2017; Zhang et al., 2017b), $L_1$ (Lim et al., 2017; Zhang et al., 2018c), perceptual and adversarial losses (Ledig et al., 2017). To show the effectiveness of our RNAN, we choose to optimize the same loss function (e.g., $L_2$ loss function) as previous works. Given a training set $\left\{I_L^i, I_H^i\right\}_{i=1}^N$, which contains $N$ low-quality inputs and their high-quality counterparts. The goal of training RNAN is to minimize the $L_2$ loss function

$$L(\Theta) = \frac{1}{N}\sum_{i=1}^N \left\|H_{RNAN}\left(I_L^i\right) - I_H^i\right\|_2,\qquad(2)$$

where $\|\cdot\|_2$ denotes $l_2$ norm. As detailed in Section 4, we use the same loss function as that in other compared methods. Such choice makes it clearer and more fair to see the effectiveness of our proposed RNAN. Then we give more details to residual local and non-local attention blocks.

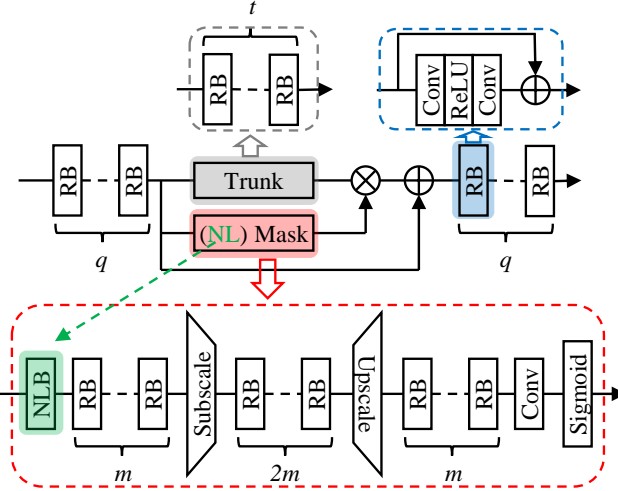

Figure 2: Residual (non-)local attention block. It mainly consists of trunk branch (labelled with gray dashed) and mask branch (labelled with red dashed). The trunk branch consists of $t$ RBs. The mask branch is used to learning mixed attention maps in channel- and spatial-wise simultaneously.

## 3.2 RESIDUAL NON-LOCAL ATTENTION BLOCK

Our residual non-local attention network is constructed by stacking several residual local and non-local attention blocks shown in Figure 2. Each attention block is divided into two parts: $q$ residual blocks (RBs) in the beginning and end of attention block. Two branches in the middle part: trunk branch and mask branch. For non-local attention block, we incorporate non-local block (NLB) in the mask branch, resulting non-local attention. Then we give more details to those components.

### 3.2.1 TRUNK BRANCH

As shown in Figure 2, the trunk branch includes $t$ residual blocks (RBs). Different from the original residual block in ResNet (He et al., 2016), we adopt the simplified RB from (Lim et al., 2017). The simplified RB (labelled with blue dashed) only consists of two convolutional layers and one ReLU (Nair & Hinton, 2010), omitting unnecessary components, such as maxpooling and batch normalization (Ioffe & Szegedy, 2015) layers. We find that such simplified RB not only contributes to image super-resolution (Lim et al., 2017), but also helps to construct very deep network for other image restoration tasks.

Feature maps from trunk branch of different depths serve as hierarchical features. If attention mechanism is not considered, the proposed network would become a simplified ResNet. With mask branch, we can take channel and spatial attention to adaptively rescale hierarchical features. Then we give more details about local and non-local attention.

### 3.2.2 MASK BRANCH

As labelled with red dashed in Figure 2, the mask branches used in our network include local and non-local ones. Here, we mainly focus on local mask branch, which can become a non-local one by using non-local block (NLB, labelled with green dashed arrow).

The key point in mask branch is how to grasp information of larger scope, namely larger receptive field size, so that it's possible to obtain more sophisticated attention map. One possible solution is to perform maxpooling several times, as used in (Wang et al., 2017) for image classification. However, more pixel-level accurate results are desired in image restoration. Maxpooling would lose lots of details of the image, resulting in bad performance. To alleviate such drawbacks, we choose to use large-stride convolution and deconvolution to enlarge receptive field size. Another way is considering non-local information across the whole inputs, which will be discussed in the next subsection.

From the input, large-stride (stride $\geq 2$) convolutional layer increases the receptive field size after $m$ RBs. After additional $2m$ RBs, the downscaled feature maps are then expanded by a deconvolutional

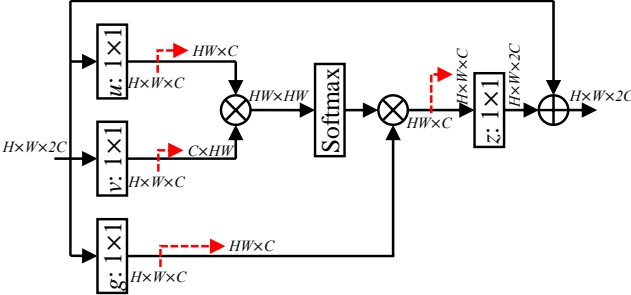

Figure 3: Non-local block. Red dashed line denotes matrix reshaping. $H \times W \times C$ means $C$ features with height $H$ and width $W$. $\otimes$ denotes matrix multiplication. $\oplus$ denotes element-wise addition.

layer (also known as transposed convolutional layer). The upscaled features are further forwarded through $m$ RBs and one $1 \times 1$ convolutional layer. Then a sigmoid layer normalizes the output values, ranging in $[0, 1]$. Although the receptive field size of the mask branch is much larger than that of the trunk branch, it cannot cover the whole features at a time. This can be achieved by using non-local block (NLB), resulting in non-local mixed attention.

### 3.2.3 NON-LOCAL MIXED ATTENTION

As discussed above, convolution operation processes one local neighbourhood at a time. In order to obtain better attention maps, here we seek to take all the positions into consideration at a time. Inspired by classical non-local means method (Buades et al., 2005) and non-local neural networks (Wang et al., 2018a), we incorporate non-local block (NLB) into the mask branch to obtain non-local mixed attention (shown in Figure 3). The non-local operation can be defined as

$$y_i = \left( \sum_{\forall j} f(x_i, x_j) g(x_j) \right) / \sum_{\forall j} f(x_i, x_j), \tag{3}$$

where $i$ is the output feature position index and $j$ is the index that enumerates all possible positions. $x$ and $y$ are the input and output of non-local operation. The pairwise function $f(x_i, x_j)$ computes relationship between $x_i$ and $x_j$. The function $g(x_j)$ computes a representation of the input at the position $j$.

As shown in Figure 3, we use embedded Gaussian function to evaluate the pairwise relationship

$$f(x_i, x_j) = exp(u(x_i)^T v(x_j)) = exp((W_u x_i)^T W_v x_j), \tag{4}$$

where $W_u$ and $W_v$ are weight matrices. As investigated in (Wang et al., 2018a), there are several versions of $f$, such as Gaussian function, dot product similarity, and feature concatenation. We also consider a linear embedding for $g$: $g(x_j) = W_g x_j$ with weight matrix $W_g$. Then the output $z$ at position $i$ of non-local block (NLB) is calculated as

$$z_i = W_z y_i + x_i = W_z softmax((W_u x_i)^T W_v x_j) g(x_j) + x_i, \tag{5}$$

where $W_z$ is a weight matrix. For a given $i$, $\sum_{\forall j} f(x_i, x_j) / \sum_{\forall j} f(x_i, x_j)$ in Eq. 3 becomes the *softmax* computation along dimension $j$. The residual connection allows us to insert the NLB into pretrained networks (Wang et al., 2018a) by initializing $W_z$ as zero.

With non-local and local attention computation, feature maps in the mask branch are finally mapped by sigmoid function

$$f_{mix}(x_{i,c}) = \frac{1}{1 + exp(-x_{i,c})}, \tag{6}$$

where $i$ ranges over spatial positions and $c$ ranges over feature channel positions. Such simple sigmoid operation is applied to each channel and spatial position, resulting mixed attention (Wang et al., 2017). As a result, the mask branch with non-local block can produce non-local mixed attention. However, simple multiplication between features from trunk and mask branches is not powerful enough or proper to form very deep trainable network. We propose residual non-local attention learning to solve those problems.

## 3.3 RESIDUAL NON-LOCAL ATTENTION LEARNING

How to train deep image restoration network with non-local mixed attention remains unclear. Here we only consider the trunk and mask branches, and residual connection with them (Figure 2). We focus on obtaining non-local attention information from the input feature $x$. It should be noted that one form of attention residual learning was proposed in Wang et al. (2017), whose formulation is

$$H_{RNA}(x) = H_{trunk}(x)(H_{mask}(x) + 1). \tag{7}$$

We find that this form of attention learning is not suitable for image restoration tasks. This is mainly because Eq. 7 is more suitable for high-level vision tasks (e.g., image classification), where low-level features are not preserved too much. However, low-level features are more important for image restoration. As a result, we propose a simple yet more suitable residual attention learning method by introducing input feature $x$ directly. We compute its output $H_{RNA}(x)$ as

$$H_{RNA}(x) = H_{trunk}(x)H_{mask}(x) + x, \tag{8}$$

where $H_{trunk}(x)$ and $H_{mask}(x)$ denote the functions of trunk and mask branches respectively. Such residual learning tends to preserve more low-level features and allows us to form very deep networks for high-quality image restoration tasks with stronger representation ability.

## 3.4 IMPLEMENTATION DETAILS

Now, we specify the implementation details of our proposed RNAN. We use 10 residual local and non-local attention blocks (2 non-local one). In each residual (non-)local block, we set $q, t, m = 2, 2, 1$. We set 3×3 as the size of all convolutional layers except for those in non-local block and convolutional layer before sigmoid function, where the kernel size is 1×1. Features in RBs have 64 filters, except for that in the non-local block (see Figure 3), where $C = 32$. In each training batch, 16 low-quality (LQ) patches with the size of $48 \times 48$ are extracted as inputs. Our model is trained by ADAM optimizer (Kingma & Ba, 2014) with $\beta_1 = 0.9$, $\beta_2 = 0.999$, and $\epsilon = 10^{-8}$. The initial learning rate is set to $10^{-4}$ and then decreases to half every $2 \times 10^5$ iterations of back-propagation. We use PyTorch (Paszke et al., 2017) to implement our models with a Titan Xp GPU.

## 4 EXPERIMENTS

We apply our proposed RNAN to three classical image restoration tasks: image denoising, demosaicing, and compression artifacts reduction. For image denoising and demosaicing, we follow the same setting as IRCNN (Zhang et al., 2017b). For image compression artifacts reduction, we follow the same setting as ARCNN (Dong et al., 2015). We use 800 training images in DIV2K (Timofte et al., 2017; Agustsson & Timofte, 2017) to train all of our models. For each task, we use commonly used dataset for testing and report PSNR and/or SSIM (Wang et al., 2004) to evaluate the results of each method. More results are shown in Appendix A.

Table 1: Ablation study of different components. PSNR values are based on Urban100 ($\sigma$=30).

| Case Index | 1 | 2 | 3 | 4 | 5 | 6 | 7 | 8 |
|---|---|---|---|---|---|---|---|---|
| Mask Branch | ✗ | ✓ | ✗ | ✓ | ✓ | ✓ | ✓ | ✓ |
| Non-local Block | ✗ | ✗ | ✓ | ✓ | ✓ | ✓ | ✓ | ✓ |
| RAB Number | 7 | 7 | 5 | 5 | 1 | 2 | 5 | 8 |
| RNAB Number | 0 | 0 | 2 | 2 | 1 | 2 | 1 | 2 |
| PSNR (dB) | 30.96 | 31.17 | 31.20 | 31.32 | 30.78 | 30.99 | 31.27 | 31.50 |

## 4.1 ABLATION STUDY

We show ablation study in Table 1 to investigate the effects of different components in RNAN.

**Non-local Mixed Attention**. In cases 1, all the mask branches and non-local blocks are removed. In case 4, we enable non-local mixed attention with same block number as in case 1. The positive effect of non-local mixed attention is demonstrated by its obvious performance improvement.

**Mask branch**. We also learn that mask branch contributes to performance improvement, no matter non-local blocks are used (cases 3 and 4) or not (cases 1 and 2). This's mainly because mask branch provides informative attention to the network, gaining better representational ability.

**Non-local block**. Non-local block also contributes to the network ability obviously, no matter mask branches are used (cases 2 and 4) or not (cases 1 and 3). With non-local information from low-level and high-level features, RNAN performs better image restoration.

**Block Number**. When comparing cases 2, 4, and 7, we learn that more non-local blocks achieve better results. However, the introduction of non-local block consumes much time. So we use 2 non-local blocks by considering low- and high-level features. When RNAB number is fixed in cases 5 and 7 or cases 6 and 8, performance also benefits from more RABs.

Table 2: Quantitative results about **color** image denoising. Best results are **highlighted**.

| Method | Kodak24 | | | | BSD68 | | | | Urban100 | | | |
|---|---|---|---|---|---|---|---|---|---|---|---|---|
| | 10 | 30 | 50 | 70 | 10 | 30 | 50 | 70 | 10 | 30 | 50 | 70 |
| CBM3D | 36.57 | 30.89 | 28.63 | 27.27 | 35.91 | 29.73 | 27.38 | 26.00 | 36.00 | 30.36 | 27.94 | 26.31 |
| TNRD | 34.33 | 28.83 | 27.17 | 24.94 | 33.36 | 27.64 | 25.96 | 23.83 | 33.60 | 27.40 | 25.52 | 22.63 |
| RED | 34.91 | 29.71 | 27.62 | 26.36 | 33.89 | 28.46 | 26.35 | 25.09 | 34.59 | 29.02 | 26.40 | 24.74 |
| DnCNN | 36.98 | 31.39 | 29.16 | 27.64 | 36.31 | 30.40 | 28.01 | 26.56 | 36.21 | 30.28 | 28.16 | 26.17 |
| MemNet | N/A | 29.67 | 27.65 | 26.40 | N/A | 28.39 | 26.33 | 25.08 | N/A | 28.93 | 26.53 | 24.93 |
| IRCNN | 36.70 | 31.24 | 28.93 | N/A | 36.06 | 30.22 | 27.86 | N/A | 35.81 | 30.28 | 27.69 | N/A |
| FFDNet | 36.81 | 31.39 | 29.10 | 27.68 | 36.14 | 30.31 | 27.96 | 26.53 | 35.77 | 30.53 | 28.05 | 26.39 |
| RNAN (ours) | **37.24** | **31.86** | **29.58** | **28.16** | **36.43** | **30.63** | **28.27** | **26.83** | **36.59** | **31.50** | **29.08** | **27.45** |

Table 3: Quantitative results about **gray-scale** image denoising. Best results are **highlighted**.

| Method | Kodak24 | | | | BSD68 | | | | Urban100 | | | |
|---|---|---|---|---|---|---|---|---|---|---|---|---|
| | 10 | 30 | 50 | 70 | 10 | 30 | 50 | 70 | 10 | 30 | 50 | 70 |
| BM3D | 34.39 | 29.13 | 26.99 | 25.73 | 33.31 | 27.76 | 25.62 | 24.44 | 34.47 | 28.75 | 25.94 | 24.27 |
| TNRD | 34.41 | 28.87 | 27.20 | 24.95 | 33.41 | 27.66 | 25.97 | 23.83 | 33.78 | 27.49 | 25.59 | 22.67 |
| RED | 35.02 | 29.77 | 27.66 | 26.39 | 33.99 | 28.50 | 26.37 | 25.10 | 34.91 | 29.18 | 26.51 | 24.82 |
| DnCNN | 34.90 | 29.62 | 27.51 | 26.08 | 33.88 | 28.36 | 26.23 | 24.90 | 34.73 | 28.88 | 26.28 | 24.36 |
| MemNet | N/A | 29.72 | 27.68 | 26.42 | N/A | 28.43 | 26.35 | 25.09 | N/A | 29.10 | 26.65 | 25.01 |
| IRCNN | 34.76 | 29.53 | 27.45 | N/A | 33.74 | 28.26 | 26.15 | N/A | 34.60 | 28.85 | 26.24 | N/A |
| FFDNet | 34.81 | 29.70 | 27.63 | 26.34 | 33.76 | 28.39 | 26.29 | 25.04 | 34.45 | 29.03 | 26.52 | 24.86 |
| RNAN (ours) | **35.20** | **30.04** | **27.93** | **26.60** | **34.04** | **28.61** | **26.48** | **25.18** | **35.52** | **30.20** | **27.65** | **25.89** |

## 4.2 COLOR AND GRAY IMAGE DENOISING

We compare our RNAN with state-of-the-art denoising methods: BM3D (Dabov et al., 2007b), CBM3D (Dabov et al., 2007a), TNRD (Chen & Pock, 2017), RED (Mao et al., 2016), DnCNN (Zhang et al., 2017a), MemNet (Tai et al., 2017), IRCNN (Zhang et al., 2017b), and FFD-Net (Zhang et al., 2017c). Kodak24 (http://r0k.us/graphics/kodak/), BSD68 (Martin et al., 2001), and Urban100 (Huang et al., 2015) are used for color and gray-scale image denoising. AWGN noises of different levels (e.g., 10, 30, 50, and 70) are added to clean images.

Quantitative results are shown in Tables 2 and 3. As we can see that our proposed RNAN achieves the best results on all the datasets with all noise levels. Our proposed non-local attention covers the information from the whole image, which should be effective for heavy image denoising. To demonstrate this analysis, we take noise level $\sigma = 70$ as an example. We can see that our proposed RNAN achieves 0.48, 0.30, and 1.06 dB PSNR gains over the second best method FFDNet. This comparison strongly shows the effectiveness of our proposed non-local mixed attention.

We also show visual results in Figures 4 and 5. With the learned non-local mixed attention, RNAN treats different image parts distinctively, alleviating over-smoothing artifacts obviously.

Table 4: Quantitative results about color image demosaicing. Best results are **highlighted**.

| Method | McMaster18 | | Kodak24 | | BSD68 | | Urban100 | |
|---|---|---|---|---|---|---|---|---|
| | PSNR | SSIM | PSNR | SSIM | PSNR | SSIM | PSNR | SSIM |
| Mosaiced | 9.17 | 0.1674 | 8.56 | 0.0682 | 8.43 | 0.0850 | 7.48 | 0.1195 |
| IRCNN | 37.47 | 0.9615 | 40.41 | 0.9807 | 39.96 | 0.9850 | 36.64 | 0.9743 |
| RNAN (ours) | **39.71** | **0.9725** | **43.09** | **0.9902** | **42.50** | **0.9929** | **39.75** | **0.9848** |

## 4.3 IMAGE DEMOSAICING

Following the same setting in IRCNN (Zhang et al., 2017b), we compare image demosaicing results with those of IRCNN on McMaster (Zhang et al., 2017b), Kodak24, BSD68, and Urban100. Since IRCNN has been one of the best methods for image demosaicing and limited space, we only compare with IRCNN in Table 4. As we can see, mosaiced images have very poor quality, resulting in very low PSNR and SSIM values. IRCNN can enhance the low-quality images and achieve relatively high values of PSNR and SSIM. Our RNAN can still make significant improvements over IRCNN. Using local and non-local attention, our RNAN can better handle the degradation situation.

Visual results are shown in Figure 6. Although IRCNN can remove mosaicing effect greatly, there're still some artifacts in its results (e.g., blocking artifacts in 'img_026'). However, RNAN recovers more faithful color and alliciates blocking artifacts.

## 4.4 Image Compression Artifacts Reduction

We further apply our RNAN to reduce image compression artifacts. We compare our RNAN with SA-DCT (Foi et al., 2007), ARCNN (Dong et al., 2015), TNRD (Chen & Pock, 2017), and DnCNN (Zhang et al., 2017a). We apply the standard JPEG compression scheme to obtain the compressed images by following (Dong et al., 2015). Four JPEG quality settings $q$ = 10, 20, 30, 40 are used in Matlab JPEG encoder. Here, we only focus on the restoration of Y channel (in YCbCr space) to keep fair comparison with other methods. We use the same datasets LIVE1 (Sheikh et al., 2005) and Classic5 (Foi et al., 2007) in ARCNN and report PSNR/SSIM values in Table 5. As we can see, our RNAN achieves the best PSNR and SSIM values on LIVE1 and Classic5 with all JPEG qualities.

We further shown visual comparisons in Figure 7. We provide comparisons under very low image quality ($q$=10). The blocking artifacts can be removed to some degree, but ARCNN, TNRD, and DnCNN would also over-smooth some structures. RNAN obtains more details with consistent structures by considering non-local mixed attention.

Table 5: Quantitative results about compression artifacts reduction. Best results are **highlighted**.

| Dataset | $q$ | JPEG | | SA-DCT | | ARCNN | | TNRD | | DnCNN | | RNAN (ours) | |
|---------|-----|------|------|--------|------|-------|------|------|------|-------|------|------|------|
| | | PSNR | SSIM | PSNR | SSIM | PSNR | SSIM | PSNR | SSIM | PSNR | SSIM | PSNR | SSIM |
| LIVE1 | 10 | 27.77 | 0.7905 | 28.65 | 0.8093 | 28.98 | 0.8217 | 29.15 | 0.8111 | 29.19 | 0.8123 | **29.63** | **0.8239** |
| | 20 | 30.07 | 0.8683 | 30.81 | 0.8781 | 31.29 | 0.8871 | 31.46 | 0.8769 | 31.59 | 0.8802 | **32.03** | **0.8877** |
| | 30 | 31.41 | 0.9000 | 32.08 | 0.9078 | 32.69 | 0.9166 | 32.84 | 0.9059 | 32.98 | 0.9090 | **33.45** | **0.9149** |
| | 40 | 32.35 | 0.9173 | 32.99 | 0.9240 | 33.63 | 0.9306 | N/A | N/A | 33.96 | 0.9247 | **34.47** | **0.9299** |
| Classic5 | 10 | 27.82 | 0.7800 | 28.88 | 0.8071 | 29.04 | 0.8111 | 29.28 | 0.7992 | 29.40 | 0.8026 | **29.96** | **0.8178** |
| | 20 | 30.12 | 0.8541 | 30.92 | 0.8663 | 31.16 | 0.8694 | 31.47 | 0.8576 | 31.63 | 0.8610 | **32.11** | **0.8693** |
| | 30 | 31.48 | 0.8844 | 32.14 | 0.8914 | 32.52 | 0.8967 | 32.78 | 0.8837 | 32.91 | 0.8861 | **33.38** | **0.8924** |
| | 40 | 32.43 | 0.9011 | 33.00 | 0.9055 | 33.34 | 0.9101 | N/A | N/A | 33.77 | 0.9003 | **34.27** | **0.9061** |

## 4.5 Image Super-Resolution

We further compare our RNAN with state-of-the-art SR methods: EDSR (Lim et al., 2017), SR-MDNF (Zhang et al., 2018a), D-DBPN (Haris et al., 2018), and RCAN (Zhang et al., 2018b). Similar to (Lim et al., 2017; Zhang et al., 2018c), we also introduce self-ensemble strategy to further improve our RNAN and denote the self-ensembled one as RNAN+.

As shown in Table 6, our RNAN+ achieves the second best performance among benchmark datasets: Set5 (Bevilacqua et al., 2012), Set14 (Zeyde et al., 2010), B100 (Martin et al., 2001), Urban100 (Huang et al., 2015), and Manga109 (Matsui et al., 2017). Even without self-ensemble, our RNAN achieves third best results in most cases. Such improvements are notable, because the parameter number of RNAN is 7.5 M, far smaller than 43 M in EDSR and 16 M in RCAN. The network depth of our RNAN (about 120 convolutional layers) is also far shallower than that of RCAN, which has about 400 convolutional layers. It indicates that non-local attention make better use of main network, saving much network parameter.

In Figure 8, we conduct image SR ($\times$4) with several state-of-the-art methods. We can see that our RNAN obtains better visually pleasing results with finer structures. These comparisons further demonstrate the effectiveness of our proposed RNAN with the usage of non-local mixed attention.

Table 6: Quantitative image SR results. Best and second best results are **highlighted** and underlined

| Method | Scale | Set5 | | Set14 | | B100 | | Urban100 | | Manga109 | |
|--------|-------|------|------|-------|------|------|------|----------|------|----------|------|
| | | PSNR | SSIM | PSNR | SSIM | PSNR | SSIM | PSNR | SSIM | PSNR | SSIM |
| Bicubic | $\times$2 | 33.66 | 0.9299 | 30.24 | 0.8688 | 29.56 | 0.8431 | 26.88 | 0.8403 | 30.80 | 0.9339 |
| EDSR | $\times$2 | 38.11 | 0.9602 | 33.92 | 0.9195 | 32.32 | 0.9013 | 32.93 | 0.9351 | 39.10 | 0.9773 |
| SRMDNF | $\times$2 | 37.79 | 0.9601 | 33.32 | 0.9159 | 32.05 | 0.8985 | 31.33 | 0.9204 | 38.07 | 0.9761 |
| D-DBPN | $\times$2 | 38.09 | 0.9600 | 33.85 | 0.9190 | 32.27 | 0.9000 | 32.55 | 0.9324 | 38.89 | 0.9775 |
| RCAN | $\times$2 | **38.27** | **0.9614** | **34.12** | 0.9216 | **32.41** | **0.9027** | **33.34** | **0.9384** | **39.44** | 0.9786 |
| RNAN (ours) | $\times$2 | 38.17 | 0.9611 | 33.87 | 0.9207 | 32.32 | 0.9014 | 32.73 | 0.9340 | 39.23 | 0.9785 |
| RNAN+ (ours) | $\times$2 | 38.22 | 0.9613 | 33.97 | **0.9216** | 32.36 | 0.9018 | 32.90 | 0.9351 | 39.41 | **0.9789** |
| Bicubic | $\times$4 | 28.42 | 0.8104 | 26.00 | 0.7027 | 25.96 | 0.6675 | 23.14 | 0.6577 | 24.89 | 0.7866 |
| EDSR | $\times$4 | 32.46 | 0.8968 | 28.80 | 0.7876 | 27.71 | 0.7420 | 26.64 | 0.8033 | 31.02 | 0.9148 |
| SRMDNF | $\times$4 | 31.96 | 0.8925 | 28.35 | 0.7787 | 27.49 | 0.7337 | 25.68 | 0.7731 | 30.09 | 0.9024 |
| D-DBPN | $\times$4 | 32.47 | 0.8980 | 28.82 | 0.7860 | 27.72 | 0.7400 | 26.38 | 0.7946 | 30.91 | 0.9137 |
| RCAN | $\times$4 | **32.63** | **0.9002** | 28.87 | **0.7889** | **27.77** | **0.7436** | **26.82** | **0.8087** | 31.22 | 0.9173 |
| RNAN (ours) | $\times$4 | 32.49 | 0.8982 | 28.83 | 0.7878 | 27.72 | 0.7421 | 26.61 | 0.8023 | 31.09 | 0.9149 |
| RNAN+ (ours) | $\times$4 | 32.56 | 0.8992 | **28.90** | 0.7883 | 27.77 | 0.7424 | 26.75 | 0.8052 | **31.37** | **0.9175** |

### 4.6 PARAMETERS AND RUNNING TIME ANALYSES

We also compare parameters, running time, and performance based on color image denoising in Table 7. PSNR values are tested on Urban100 ($\sigma$=50). RNAN with 10 blocks achieves the best performance with the highest parameter number, which can be reduced to only 2 blocks and obtains second best performance. Here, we report running time for reference, because the time is related to implementation platform and code.

Table 7: Parameter and time comparisons. '*' means being applied channel-wise for color images.

| Methods | RED* | DnCNN | MemNet* | RNAN (1LB+1NLB) | RNAN (8LBs+2NLBs) |
|---|---|---|---|---|---|
| Parameter Number | 4,131 K | 672K | 677K | 1,494k | 7,409K |
| PSNR (dB) | 26.40 | 28.16 | 26.53 | 28.36 | 29.15 |
| Time (s) (Platform) | 58.7 (Caffe) | 16.5 (Matlab+GPU Array) | 239.0 (Caffe) | 2.6 (PyTorch) | 6.5 (PyTorch) |

## 5 CONCLUSIONS

In this paper, we propose residual non-local attention networks for high-quality image restoration. The networks are built by stacking local and non-local attention blocks, which extract local and non-local attention-aware features and consist of trunk and (non-)local mask branches. They're used to extract hierarchical features and adaptively rescale hierarchical features with soft weights. We further generate non-local attention by considering the whole feature map. Furthermore, we propose residual local and non-local attention learning to train very deep networks. We introduce the input feature into attention computation, being more suitable for image restoration. RNAN achieves state-of-the-art image restoration results with moderate model size and running time.

**Acknowledgements**. This research is supported in part by the NSF IIS award 1651902 and U.S. Army Research Office Award W911NF-17-1-0367.

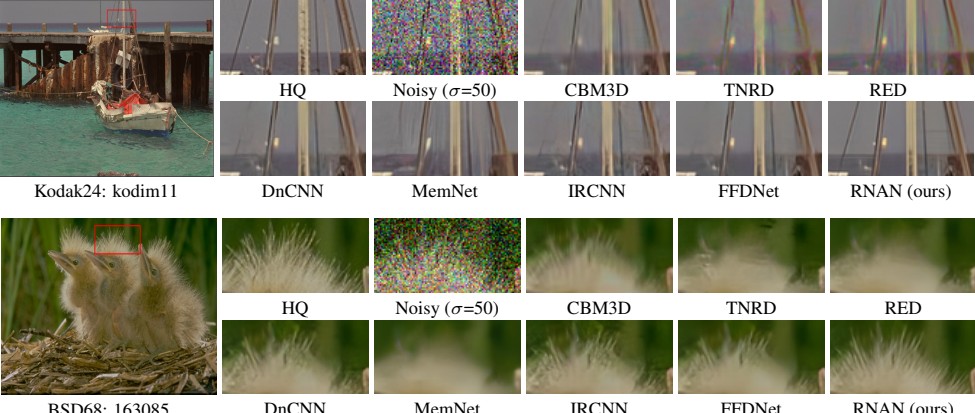

Figure 4: **Color** image denoising results with noise level $\sigma = 50$.

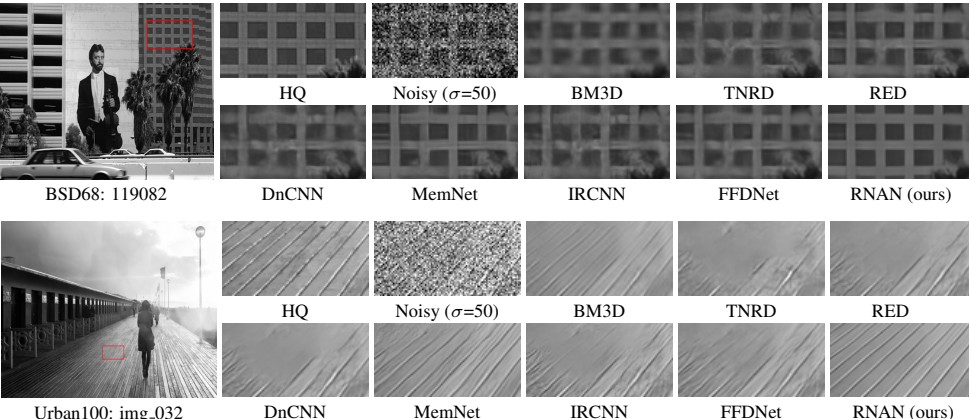

Figure 5: **Gray** image denoising results with noise level $\sigma = 50$.

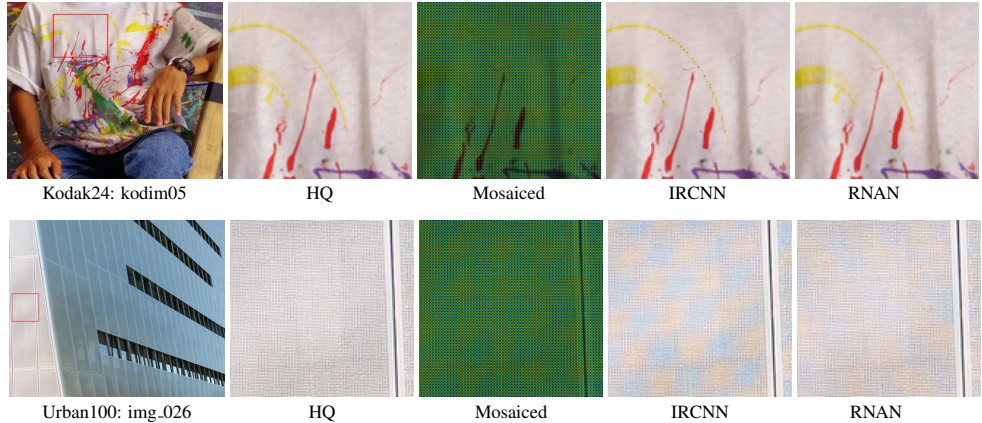

Figure 6: Image demosaicing results.

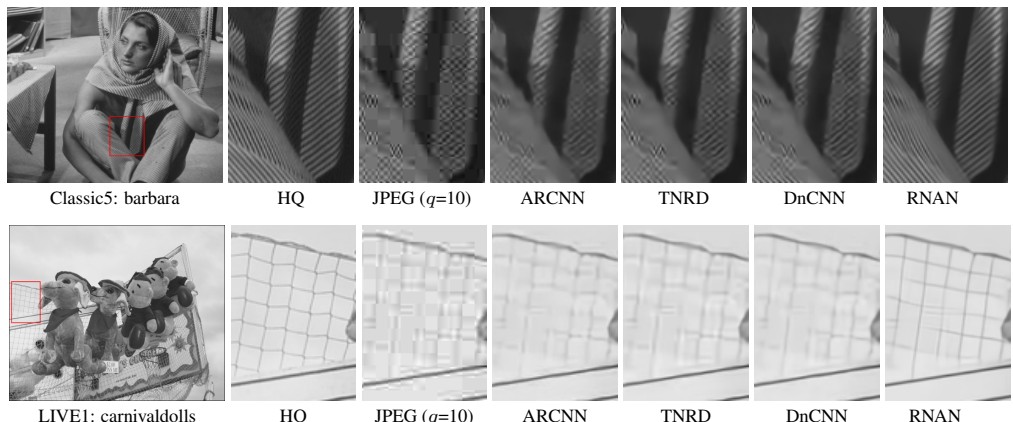

Figure 7: Image compression artifacts reduction results with JPEG quality $q = 10$.

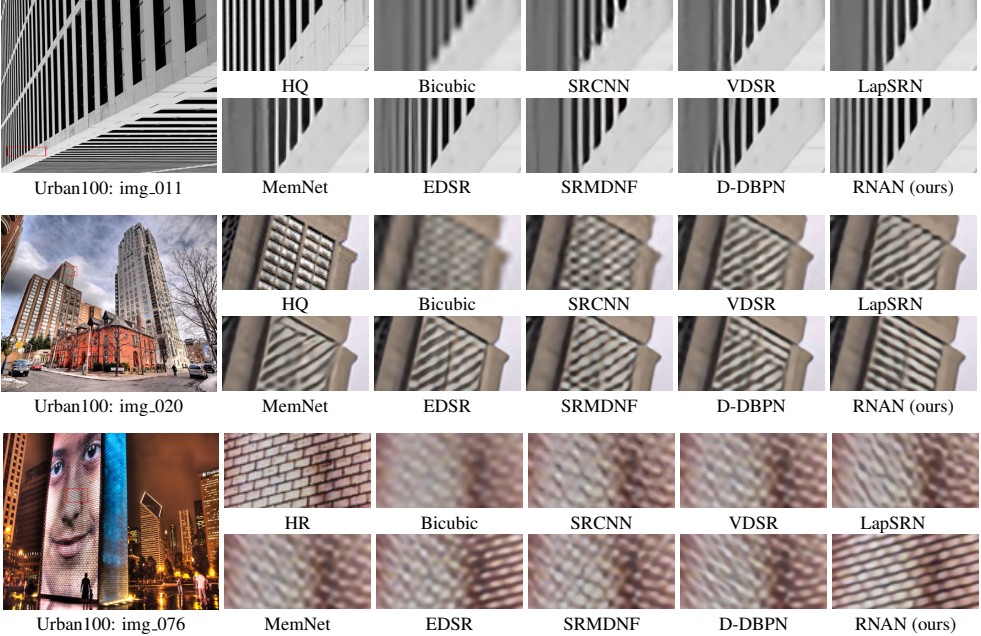

Figure 8: Image super-resolution results with scaling factor $s = 4$.

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

# A APPENDIX

## A.1 TRAIN RNAN WITH SMALL TRAINING DATA

All the results shown in the main paper are based on DIV2K training data. Here, we retrain our RNAN on small training sets for 5 tasks. In Table 8, for each task, we only refer the second best method from our main paper to compare. For training data, we use BSD400 (Martin et al., 2001) for color/gray-scale image denoising and demosaicing. We use 91 images in Yang et al. (2008) and 200 images in Martin et al. (2001) (denoted as SR291) for image compression artifacts reduction and super-resolution. FFDNet used BSD400 (Martin et al., 2001), 400 images from ImageNet Deng et al. (2009), and 4,744 images in Waterloo Exploration Database Ma et al. (2017). Here, 'BSD400+' is used to denote 'BSD400+ImageNet400+WED4744'. According to Table 8, we use the same or even smaller training set for our RNAN and obtain better results for 5 tasks. These experiments demonstrate the effectiveness of our RNAN for general image restoration tasks.

Table 8: Quantitative results about image restoration. Best results are **highlighted**.

| Task 1: Color image denoising. $\sigma$ denotes noise level. | | | | | | | | | | | | | | |
| --- | --- | --- | --- | --- | --- | --- | --- | --- | --- | --- | --- | --- | --- |
| Data | Method | Kodak24 | | | | BSD68 | | | | Urban100 | | | |
| | | $\sigma$=10 | $\sigma$=30 | $\sigma$=50 | $\sigma$=70 | $\sigma$=10 | $\sigma$=30 | $\sigma$=50 | $\sigma$=70 | $\sigma$=10 | $\sigma$=30 | $\sigma$=50 | $\sigma$=70 |
| BSD400+ | FFDNet | 36.81 | 31.39 | 29.10 | 27.68 | 36.14 | 30.31 | 27.96 | 26.53 | 35.77 | 30.53 | 28.05 | 26.39 |
| BSD400 | RNAN | **37.16** | **31.81** | **29.55** | **28.15** | **36.60** | **30.73** | **28.35** | **26.88** | **36.39** | **30.90** | **28.65** | **27.11** |

| Task 2: **Gray** image denoising. $\sigma$ denotes noise level. | | | | | | | | | | | | | | |
| --- | --- | --- | --- | --- | --- | --- | --- | --- | --- | --- | --- | --- | --- |
| Data | Method | Kodak24 | | | | BSD68 | | | | Urban100 | | | |
| | | $\sigma$=10 | $\sigma$=30 | $\sigma$=50 | $\sigma$=70 | $\sigma$=10 | $\sigma$=30 | $\sigma$=50 | $\sigma$=70 | $\sigma$=10 | $\sigma$=30 | $\sigma$=50 | $\sigma$=70 |
| BSD400+ | FFDNet | 34.81 | 29.70 | 27.63 | 26.34 | 33.76 | 28.39 | 26.29 | 25.04 | 34.45 | 29.03 | 26.52 | 24.86 |
| BSD400 | RNAN | **35.15** | **29.87** | **27.92** | **26.50** | **34.18** | **28.72** | **26.61** | **25.34** | **35.08** | **29.82** | **27.30** | **25.57** |

| Task 3: Image demosaicing | | | | Task 4: Image compression artifacts reduction. $q$ denotes JPEG quality. | | | | | |
| --- | --- | --- | --- | --- | --- | --- | --- | --- | --- |
| Data | Method | Kodak24 | BSD68 | Data | | Method | LIVE1 | | Classic5 | |
| | | | | | | | $q$=20 | $q$=40 | $q$=20 | $q$=40 |
| BSD400 | IRCNN | 40.41 | 39.96 | SR291 | | DnCNN | 31.59 | 33.96 | 31.63 | 33.77 |
| BSD400 | RNAN | **42.86** | **42.61** | SR291 | | RNAN | **31.87** | **34.32** | **31.99** | **34.03** |

| Task 5: Image super-resolution (SR). Low-resolution (LR) images are obtained by Bicubic downsampling. | | | | | | | | | | | | | | |
| --- | --- | --- | --- | --- | --- | --- | --- | --- | --- | --- | --- | --- | --- |
| Data | Method | Set5 | | | Set14 | | | B100 | | | Urban100 | | |
| | | ×2 | ×3 | ×4 | ×2 | ×3 | ×4 | ×2 | ×3 | ×4 | ×2 | ×3 | ×4 |
| SR291 | MemNet | 37.78 | 34.09 | 31.74 | 33.28 | 30.00 | 28.26 | 32.08 | 28.96 | 27.40 | 31.31 | 27.56 | 25.50 |
| SR291 | RNAN | **37.95** | **34.30** | **31.93** | **33.41** | **30.22** | **28.43** | **32.23** | **29.09** | **27.51** | **31.64** | **27.84** | **25.71** |

## A.2 VISUAL RESULTS

**Color and Gray Image Denoising**. We show color and gray-scale image denoising comparisons in Figures 9 and 10 respectively. We can see that our RNAN recovers shaper edges. Unlike most of other methods, which over-smooth some details (e.g., tiny lines), RNAN can reduce noise and maintain more details. With the learned non-local mixed attention, RNAN treats different image parts distinctively, alleviating over-smoothing artifacts obviously.

**Image Compression Artifacts Reduction (CAR)**. In Figure 11, we provide comparisons under very low image quality ($q$=10). The blocking artifacts can be removed to some degree, but ARCNN, TNRD, and DnCNN would also over-smooth some structures. In contrast, RNAN obatins more details with consistent structures by considering non-local mixed attention.

**Image Super-Resolution (SR)**. In Figure 12, we conduct image SR (×4) with several state-of-the-art methods, such as SRCNN (Dong et al., 2016), VDSR (Kim et al., 2016), LapSRN (Lai et al., 2017), MemNet (Tai et al., 2017), EDSR (Lim et al., 2017), SRMDNF (Zhang et al., 2018a), and D-DBPN (Haris et al., 2018). We can see that most of compared methods would suffer from distortion or output totally wrong structures. For tiny line, edge structures, or some textures, they fail to recover and introduce blurring artifacts. Instead, our RNAN obtains better visually pleasing results with finer structures. These comparisons further demonstrate the effectiveness of our proposed RNAN with the usage of non-local mixed attention.

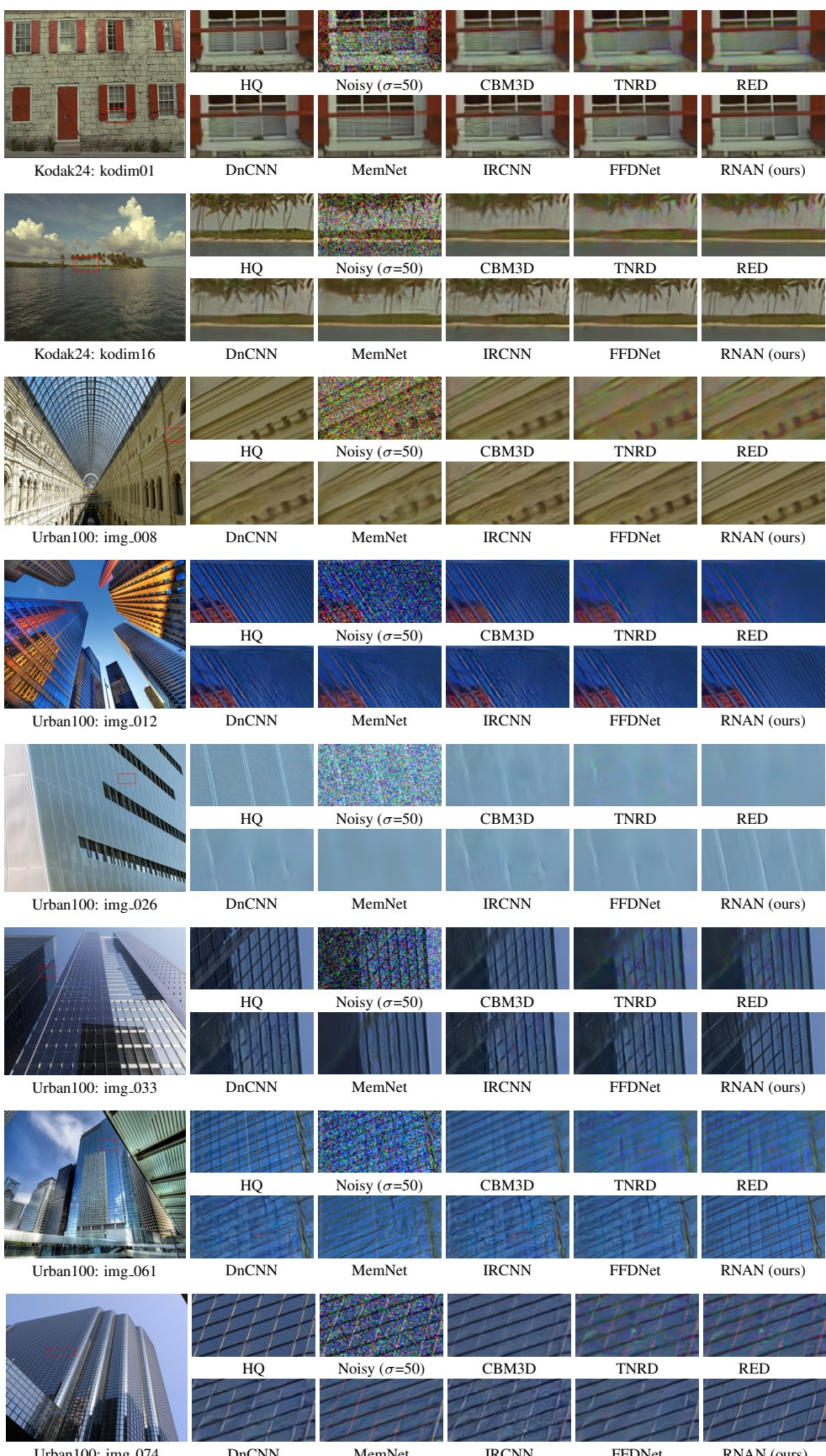

Figure 9: **Color** image denoising results with noise level $\sigma = 50$.

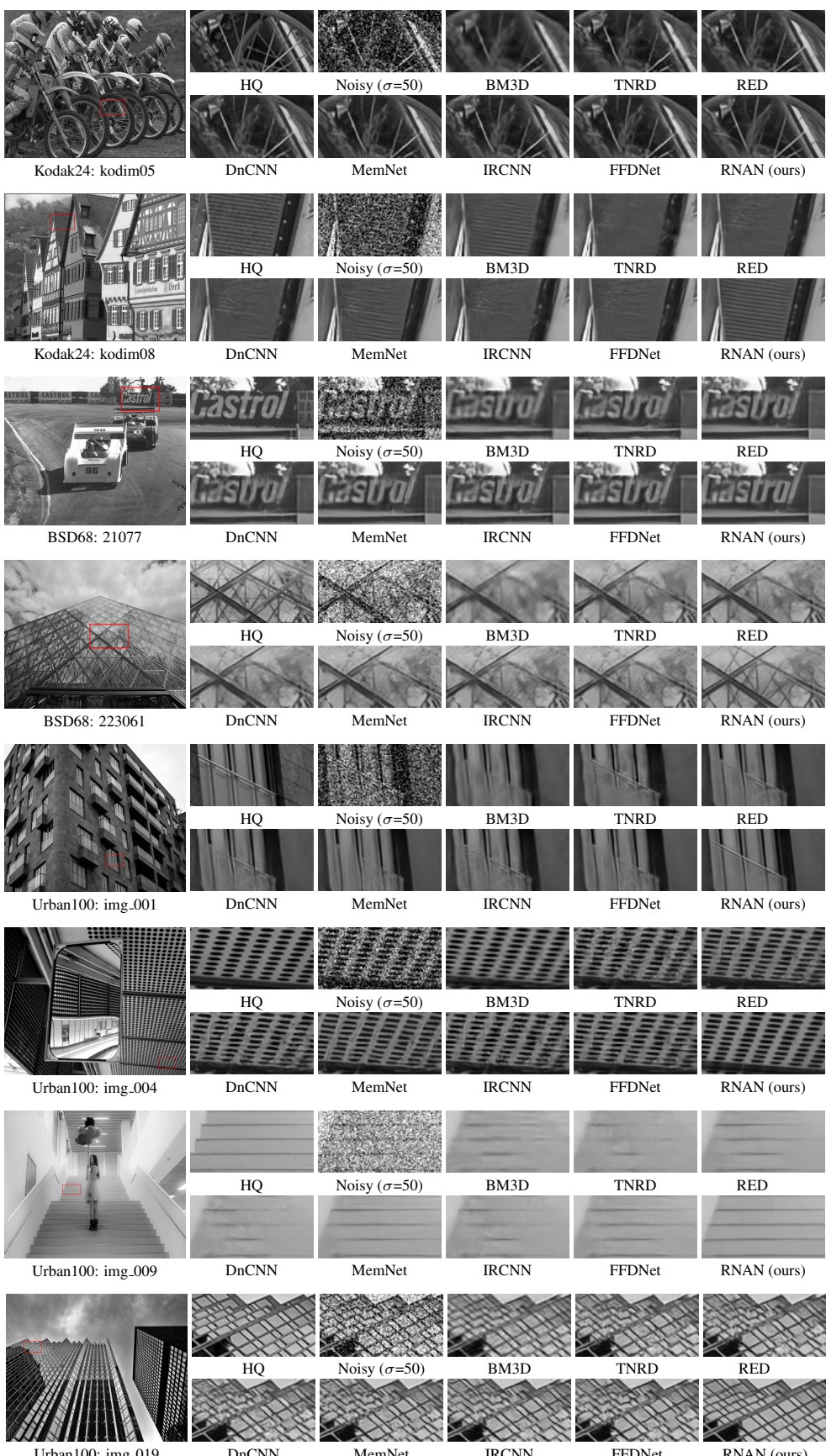

Figure 10: **Gray-scale** image denoising results with noise level $\sigma = 50$.

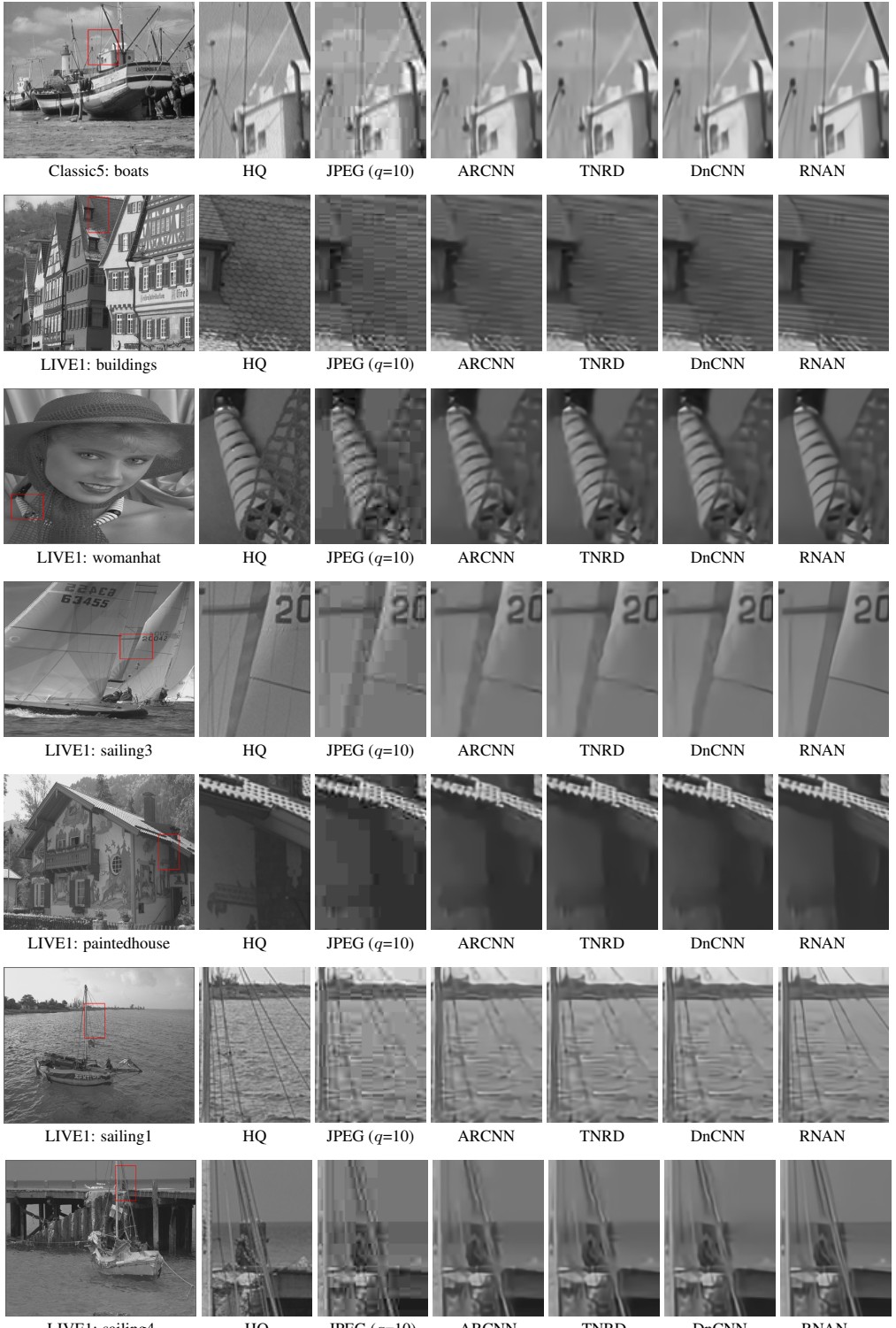

Figure 11: Image compression artifacts reduction results with JPEG quality $q = 10$.

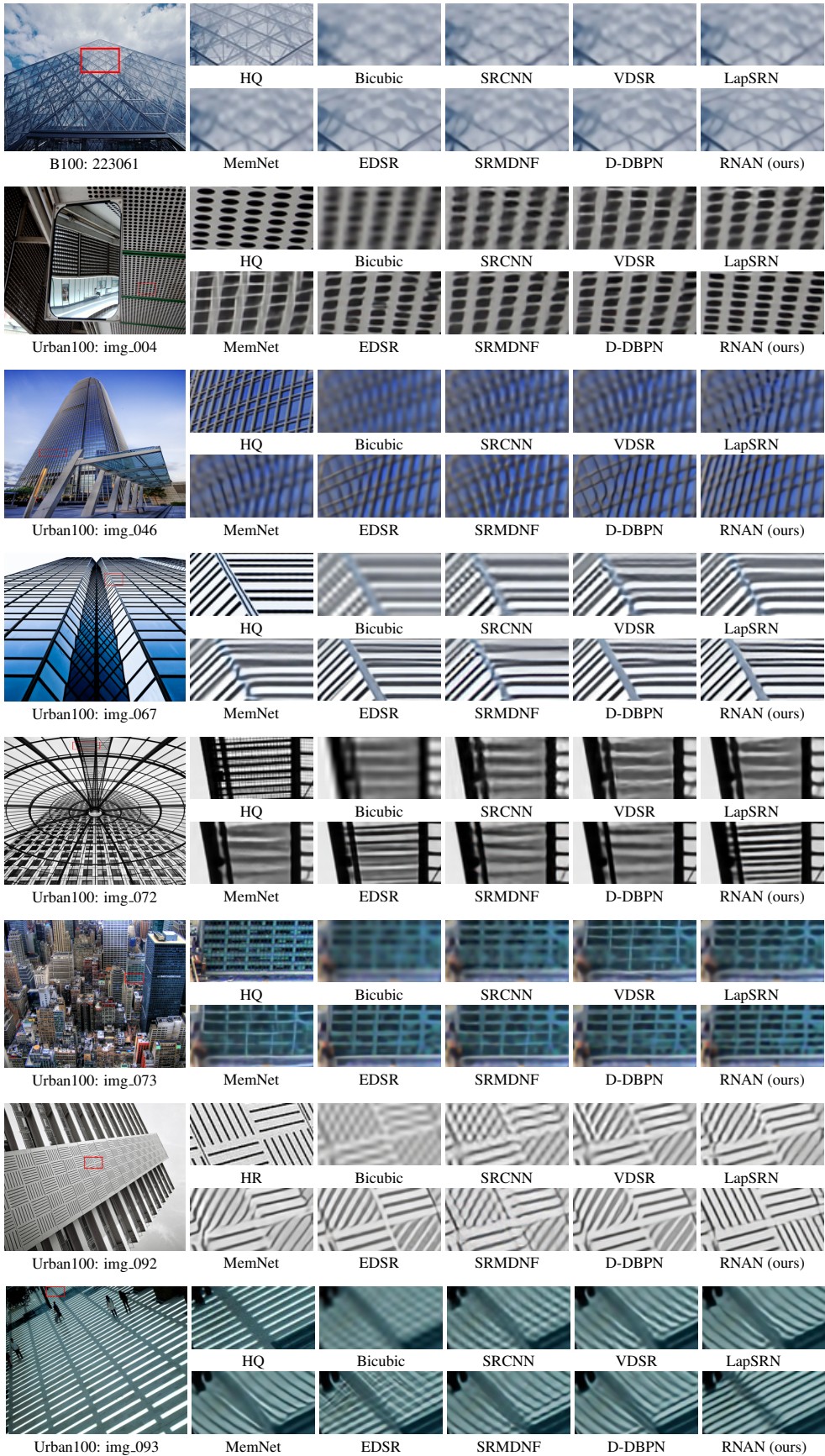

Figure 12: Image super-resolution results with scaling factor $s = 4$.

