# OpenReview forum: "Residual Non-local Attention Networks for Image Restoration"
_ICLR.cc/2019/Conference_

### Official Review · AnonReviewer1 · 2018-11-02
**Technical contribution is not high, but good performing approach on several image restoration tasks**

**Rating:** 6
**Confidence:** 3

**Review:**

- Summary
This paper proposes a residual non-local attention network for image restoration. Specifically, the proposed method has local and non-local attention blocks to extract features which capture long-range dependencies. The local and non-local blocks consist of trunk branch and (non-) local mask branch. The proposed method is evaluated on image denoising, demosaicing, compression artifacts reduction, and super-resolution.

- Pros
  - The proposed method shows better performance than existing image restoration methods.
  - The effect of each proposed technique such as the mask branch and the non-local block is appropriately evaluated.

- Cons
  - It would be better to provide the state-of-the-art method[1] in the super-resolution task.
    [1] Y. Zhang et al., Image Super-Resolution Using Very Deep Residual Channel Attention Networks, ECCV, 2018.
  - The technical contribution of the proposed method is not high, because the proposed method seems to be just using existing methods.
  - The contribution of the non-local operation is not clear to me. For example, how does the global information (i.e., long-range dependencies between pixels) help to solve image denoising tasks such as image denoising?

Overall, the technical contribution of the proposed method is not so high, but the proposed method is valuable and promising if we focus on the performance.

---

> ### Author Response · Authors · 2018-11-26
> **Author response to Reviewer1 (part 1 of 2)**
>
> We thank Reviewer1 for his/her valuable comments. We will release the code and pretrained model reproducing the results in the paper soon. Our responses are as follows:
>
> Q1-1: - Cons
>   - It would be better to provide the state-of-the-art method[1] in the super-resolution task.
>     [1] Y. Zhang et al., Image Super-Resolution Using Very Deep Residual Channel Attention Networks, ECCV, 2018.
> A1-1: Thanks for the suggestion. RCAN [1] is very powerful and shows great performance gains over previous SR methods. We include the RCAN [1] for comparison in the revised paper. It should be noted that RCAN mainly focus on much deeper network design and channel attention. Our network depth is much shallower than that of RCAN. Our RNAN mainly focus on investigating residual non-local attention and its application for image restoration. We believe that our RNAN could also contribute to RCAN to obtain better performance.
>
> Q1-2: - The technical contribution of the proposed method is not high, because the proposed method seems to be just using existing methods.
> A1-2: Our main principle of network design is to make it ‘Compact yet work’. This work mainly focuses on investigating the usage of residual local and non-local attention for image restoration. Based on some existing concepts (e.g., residual block, non-local network), we conduct extensive experiments to obtain such a compact network structure and demonstrate its effectiveness. We mainly show the effectiveness of our idea and don’t pursue higher performance by further refining the network modules. We believe that more and more related works could be done to further improve such a compact network.

---

> ### Author Response · Authors · 2018-11-26
> **Author response to Reviewer1 (part 2 of 2)**
>
> Q1-3: - The contribution of the non-local operation is not clear to me. For example, how does the global information (i.e., long-range dependencies between pixels) help to solve image denoising tasks such as image denoising?
> A1-3: Zhang et al. [R2] investigated that larger patch size contributes more for image denoising to make better use of receptive field size, especially when the noise level is high. Similar observation could also be found in image super-resolution [R3]. Although large patch size makes better use of larger receptive field size, previous methods are restricted by local convolutional operation and equal treatment of spatial and channel-wise features.
> In this paper, we use non-local attention to make full use of all the pixels of the inputs simultaneously. Namely, all the positions are considered to obtain better attention maps. Such non-local mixed attention enhances the network with distinguished power for noise and image content. For example, to denoise the kodim11 in Fig. 4, all the previous methods cannot recover the line above the boat. They take the tiny line as a part of plain sky and just remove it. However, our RNAN could keep the line and remove the noise by distinctively treat the line and sky with non-local mixed attention.
> [R2] Zhang, Kai, et al. "Beyond a gaussian denoiser: Residual learning of deep cnn for image denoising." TIP 2017.
> [R3] Wang, Xintao, et al. "ESRGAN: Enhanced super-resolution generative adversarial networks." ECCVW  2018.
>
> Q1-4: Overall, the technical contribution of the proposed method is not so high, but the proposed method is valuable and promising if we focus on the performance.
> A1-4: As Reviewer2 said ‘However, up to some point all the new ConvNet designs can be seen as incremental developments of the older ones, yet they are needed for the progress of the field.’, we have to admit that too many CNN based works focus on performance. What’s more, some works by famous companies need hundreds and thousands of high-performance GPUs, use tons of data, and take tens of days to train their networks. Although they achieve very impressive results based on existing network structures, researchers (e.g., students in most universities) without so much resource cannot even run their released codes. Such works consumes so much resource that it becomes undoable for researchers with limited resources. However, such kinds of works are not challenged or blamed with their ‘novelty’ very much and there tends to be more and more such very-large-resource-consuming works.
> In contrast, in this work, we design a compact yet effective network for image restoration. We conduct extensive experiments to demonstrate the positive contributions of each component and the effectiveness of the idea. We are the first to investigate non-local attention in image restoration tasks. Although we can make more complex network structures to achieve more ‘novelty’ and better performance, we didn’t. In fact, in our later works, we obtained much better results based on the idea in this paper.
> We want to inspire other researchers to investigate more about non-local attention for the large community, image restoration, with limited resource. All the experiments can be done with one regular GPU (e.g., 12G memory). The results are also reproducible, as we will release the train/test codes and pretrained models.

---

### Official Review · AnonReviewer3 · 2018-11-04
**excellent results, but unclear novelty and lacking explanations**

**Rating:** 7
**Confidence:** 3

**Review:**

The paper proposes a convolutional neural network architecture that includes blocks for local and non-local attention mechanisms, which are claimed to be responsible for achieving excellent results in four image restoration applications.


# Results
The strongest point of the paper is that the quantitative and qualitative image restoration results appear to be very good, although they seem almost a bit too good.


# Novelty
I'm not sure about the novelty of the paper, but I suspect it to be rather incremental. The paper says "To the best of our knowledge, this is the first time to consider residual non-local attention for image restoration problems." Does that mean non-local attention (in a very similar way) has already been used, just not in a residual fashion? If so, that would not constitute much novelty. I have to admit that I'm not familiar with the related work on attention, but I did not understand *why* the results of the proposed method are supposed to be much better than that of previous work.


# Clarity
I think the paper is not self-contained enough, since it seems to implicitly assume substantial background knowledge on attention mechanisms in CNNs.

Furthermore, the introduction of the paper identifies three problems with existing CNNs that I don't necessarily fully agree with. None of these supposed problems are backed up by (experimental) evidence.

I don't think it is sufficient to just show superior results than previous methods. It is also important to disentangle why the results are better. However, the presented ablation experiments are not very illuminating to me.

The attempts at explaining what the novel attention blocks do and why they lead to superior results are very vague to me. Maybe they are understandable in the context of related work, but I found many statements, such as the following, devoid of meaning:
- "Without considering the uneven distribution of information in the corrupted images, [...]"
- "However, in this paper, we mainly focus on learning non-local attention to better guide feature extraction in trunk branch."
- "We only incorporate residual non-local attention block in low-level and high-level feature space. This is mainly because a few non-local modules can well offer non-local ability to the network for image restoration."
- "The key point in mask branch is how to grasp information of larger scope, namely larger receptive field size, so that it’s possible to obtain more sophisticated attention map."


# Experiments
- The experimental results are the best part of the paper. However, it would've been nice to include some qualitative results in the main paper.
- The proposed RNAN model is trained on a big dataset (800 images with ~2 million pixels each). Are the competing methods trained on datasets of similar size? If not, this could be a major reason for improved performance of RNAN over competing methods. At least in the appendix, RNAN and FFDNet are compared more fairly since they are trained with the same/similar data.
- The qualitative examples in the appendix mostly show close-ups/details of very structured regions (mostly stripy patterns). Please also show some other regions without self-similar structures.


# Misc
- Residual non-local attention learning (section 3.3) was not clear to me.
- The word "trunk" is used without definition or explanation.
- Fig. 2 caption is too short, please expand.

# Update (2018-11-29)
Given the substantial author feedback, I'm willing to raise my score.

---

> ### Author Response · Authors · 2018-11-26
> **Author response to Reviewer3 (part 1 of 3)**
>
> We thank Reviewer3 for his/her valuable comments. We will release the code and pretrained model reproducing the results in the paper soon. Our responses are as follows:
>
> Q3-1: # Results
> The strongest point of the paper is that the quantitative and qualitative image restoration results appear to be very good, although they seem almost a bit too good.
> A3-1: We mainly show the effectiveness of our idea and don’t pursue higher performance. We were surprising to find that our current model has achieved much better performance than most previous methods in image restoration. Actually, in our later research, we further obtained better results based on the idea in this paper. Anyway, we will release the train/test codes and pretrained models soon, which reproduce the exact results in this paper.
>
> Q3-2: # Novelty
> I'm not sure about the novelty of the paper, but I suspect it to be rather incremental. The paper says "To the best of our knowledge, this is the first time to consider residual non-local attention for image restoration problems." Does that mean non-local attention (in a very similar way) has already been used, just not in a residual fashion? If so, that would not constitute much novelty. I have to admit that I'm not familiar with the related work on attention, but I did not understand *why* the results of the proposed method are supposed to be much better than that of previous work.
> A3-2: Non-local attention was NOT used for image restoration in terms of papers in CVPR/ICCV/ECCV/NIPS/ICML/ICLR. We are the first to investigate non-local attention for image denoising, demosaicing, compression artifact reduction, and super-resolution simultaneously. The reasons why we propose residual non-local attention learning (in Section 3.3 of the main paper) are mainly as follows:
> (1) It is a proper way to incorporate non-local attention into the network and contribute to the image restoration performance.
> (2) It allows us to train very deep networks by preserving more low-level features, being more suitable for image restoration.
> (3) It allows the network to pursue better representational ability. We demonstrate its effectiveness in both the main paper and our response to Reviewer2.
> The reasons why our proposed method achieves much better results than that of previous works are as follows:
> (1) Our residual non-local attention network is an effective network structure for high-quality image restoration. No matter we use small training data (e.g., Table 8 in main paper) or DIV2K (e.g., Table 6 in the main paper), our method achieves better results than most compared ones. Let’s take image super-resolution as an example, even though some other methods have larger number of network parameters (e.g., EDSR and D-DBPN), our method still achieves better performance.
> (2) Our proposed residual attention learning allows we train very deep network, achieve stronger representation ability. We’re the first to investigate such a deep network for image denoising, demosiacing, and compression artifacts reduction.
> (3) Our proposed method is powerful enough to further take advantage of larger training set (e.g., DIV2K). As we show Table 8 in the main paper, for small training data, we only train our network about 2 hours, being far away from well-trained. While, other compared methods would have to take much longer training time. For example, MemNet (Tai et al., 2017) trains for about 5 days, almost 60 times longer than ours.

---

> ### Author Response · Authors · 2018-11-26
> **Author response to Reviewer3 (part 2 of 3)**
>
> Q3-3: # Clarity
> I think the paper is not self-contained enough, since it seems to implicitly assume substantial background knowledge on attention mechanisms in CNNs.
> A3-3: Due to the limited space, we only included key references about attention mechanisms in the previous paper. Thanks for the reviewer’s suggestions, in the revised paper, we add more descriptions about attention mechanisms.
>
> Q3-4: Furthermore, the introduction of the paper identifies three problems with existing CNNs that I don't necessarily fully agree with. None of these supposed problems are backed up by (experimental) evidence.
> A3-4: For the first issue, Zhang et al. [R2] investigated that larger patch size contributes more for image denoising to make better use of receptive field size, especially when the noise level is high. In this paper, we use non-local attention to make full use of all the pixels of the inputs simultaneously. We compared DnCNN in [R2] to show the effectiveness of our method.
> For the second issue, we provide analyses about previous methods, which didn’t use non-local attention for image restoration and lacked discriminative ability according to the specific noisy content. We also provide visual results to demonstrate our analyses. For example, to denoise the kodim11 in Fig. 4, all the previous methods cannot recover the line above the boat. They take the tiny line as a part of plain sky and just remove it. However, our RNAN could keep the line and remove the noise by distinctively treat the line and sky.
> For the third issue, previous methods seldomly take the features distinctively in channel-wise or spatial-wise. Namely, they take the feature maps equally, which lacks flexibility in the real cases. Instead, we learn non-local mixed attention to guide the network training and obtain stronger representational ability. We support this claim with the ablation study and comparisons with other methods quantitatively and qualitatively.
> [R2] Zhang, Kai, et al. "Beyond a gaussian denoiser: Residual learning of deep cnn for image denoising." TIP 2017.
>
> Q3-5: I don't think it is sufficient to just show superior results than previous methods. It is also important to disentangle why the results are better. However, the presented ablation experiments are not very illuminating to me.
> A3-5: Please refer to A3-2 for the reasons and analyses why our results are better. On the other hand, the ablation study is used to verify the effects of each proposed component. It also serves as a guidance for us to decide the final network structure.
>
> Q3-6: The attempts at explaining what the novel attention blocks do and why they lead to superior results are very vague to me. Maybe they are understandable in the context of related work, but I found many statements, such as the following, devoid of meaning:
> - "Without considering the uneven distribution of information in the corrupted images, [...]"
> - "However, in this paper, we mainly focus on learning non-local attention to better guide feature extraction in trunk branch."
> - "We only incorporate residual non-local attention block in low-level and high-level feature space. This is mainly because a few non-local modules can well offer non-local ability to the network for image restoration."
> - "The key point in mask branch is how to grasp information of larger scope, namely larger receptive field size, so that it’s possible to obtain more sophisticated attention map."
> A3-6: We summarize our main contribution as three-fold and corresponding brief explanations at the end of Introduction. We also try our best to revise the, aiming to make it better understandable to readers.
>
> Q3-7: # Experiments
> - The experimental results are the best part of the paper. However, it would've been nice to include some qualitative results in the main paper.
> A3-7: Due to the limited space, we didn’t include some qualitative results in the main body of the paper. Thanks for the reviewer’s suggestion, we add some qualitative results in the main body of the revised one.

---

> ### Author Response · Authors · 2018-11-26
> **Author response to Reviewer3 (part 3 of 3)**
>
> Q3-8: - The proposed RNAN model is trained on a big dataset (800 images with ~2 million pixels each). Are the competing methods trained on datasets of similar size? If not, this could be a major reason for improved performance of RNAN over competing methods. At least in the appendix, RNAN and FFDNet are compared more fairly since they are trained with the same/similar data.
> A3-8: First, for image super-resolution, EDSR and our RNAN used DIV2K 800 images for training. SRMDNF and D-DBPN used DIV2K 800 images and Flickr2K 2650 images for training, much more images than ours. Our ANAN obtains better results, while using similar or smaller training set and much less network parameters than those of EDSR and D-DBPN.
> Second, for image denoising, demosaicing, and compression artifacts reduction, the compared methods use smaller training size. It’s hard to use their official released code to retrain their models with DIV2K 800 images mainly for two reasons. One is that it’s very hard to preprocess data with their codes for DIV2K training data. Second, some of the compared methods (e.g., MemNet) would need large-memory GPU (e.g., Nvidia P40 with 24G memory to train MemNet) and very long training time (e.g., 5 days to train MemNet).
> However, to make fair comparisons, we retrain our RNAN with smaller dataset and show the results in Table 8. As we can see, our RNAN still achieves better results, even using smaller training data (e.g., for denoising, we use BSD400, FFDNet uses BSD400+, which has 5144 more images than BSD400). It should also be noted that we only train our network about 2 hours, being far away from well-trained. While, other compared methods would have to take much longer training time. For example, MemNet trains for about 5 days, almost 60 times longer than ours.
>
> Q3-9: - The qualitative examples in the appendix mostly show close-ups/details of very structured regions (mostly stripy patterns). Please also show some other regions without self-similar structures.
> A3-9: First, our RNAN obtains pretty good results for regions with self-similar structures. This comparison also demonstrates the effectiveness of our proposed residual non-local attention network. Thanks for the suggestions, we further add more qualitative results without self-similar structures in the revised paper.
>
> Q3-10: # Misc
> - Residual non-local attention learning (section 3.3) was not clear to me.
> - The word "trunk" is used without definition or explanation.
> - Fig. 2 caption is too short, please expand.
> A3-10: Thanks for pointing them out. We have revised the paper to make it better understandable and easy to follow. The word “trunk” mainly means main body to extract features, just being distinguished with mask branch. Moreover, we show it in the Fig. 2. We also expand the caption of Fig. 2.

---

### Official Review · AnonReviewer2 · 2018-11-06
**excellent application oriented paper; new state-of-the-art results; yet limited novelty**

**Rating:** 7
**Confidence:** 5

**Review:**

The authors propose a residual non-local attention net (RNAN) which combines local and non-local blocks to form a deep CNN architecture with application to image restoration.

The paper has a compact description, provides sufficient details, and including the appendix has an excellent experimental validation.

The proposed approach provides top results on several image restoration tasks:  image denoising, demosaicing, compression artifacts reduction, and single image super-resolution.

The main weakness of the paper is the limited novelty, as the proposed design builds upon existing ideas and concepts. However, up to some point all the new ConvNet designs can be seen as incremental developments of the older ones, yet they are needed for the progress of the field.

I would suggest to the authors the inclusion of related works such as:
Timofte et al., "NTIRE 2018 Challenge on Single Image Super-Resolution: Methods and Results", CVPRW 2018
Wang et al., "A fully progressive approach to single-image super-resolution", CVPRW 2018
Note that DIV2K dataset was introduced in:
Agustsson et al., NTIRE 2017 Challenge on Single Image Super-Resolution: Dataset and Study, CVPRW 2017

also, the more recent related works:
Blau et al., "2018 PIRM Challenge on Perceptual Image Super-resolution", ECCVW 2018
Zhang et al., "Image Super-Resolution Using Very Deep Residual Channel Attention Networks", ECCV 2018

Also, I would like a response from the authors on the following:
Why not using dilated convolutions instead of or complementary with the mask branch or other design choices from this paper?

---

> ### Author Response · Authors · 2018-11-26
> **Author response to Reviewer2**
>
> We thank Reviewer2 for his/her valuable comments and approval for our work. We will release the code and pretrained model reproducing the results in the paper soon. Our responses are as follows:
>
> Q2-1: The main weakness of the paper is the limited novelty, as the proposed design builds upon existing ideas and concepts. However, up to some point all the new ConvNet designs can be seen as incremental developments of the older ones, yet they are needed for the progress of the field.
> A2-1: Our main principle of network design is to make it ‘Compact yet work’. This work mainly focuses on investigating the usage of residual local and non-local attention for image restoration. Based on some existing concepts (e.g., residual block, non-local network), we conduct extensive experiments to obtain such a compact network structure and demonstrate its effectiveness. We mainly show the effectiveness of our idea and don’t pursue higher performance by further refining the network modules. We believe that more and more related works could be done to further improve such a compact network.
>
> Q2-2: Inclusion of more related works, such as:
> Timofte et al., "NTIRE 2018 Challenge on Single Image Super-Resolution: Methods and Results", CVPRW 2018
> Wang et al., "A fully progressive approach to single-image super-resolution", CVPRW 2018
> Agustsson et al., NTIRE 2017 Challenge on Single Image Super-Resolution: Dataset and Study, CVPRW 2017
> Blau et al., "2018 PIRM Challenge on Perceptual Image Super-resolution", ECCVW 2018
> Zhang et al., "Image Super-Resolution Using Very Deep Residual Channel Attention Networks", ECCV 2018
> A2-2: The NTIRE and PIRM challenges and recent related works really contribute to the image restoration community very much. We have included those valuable works and given corresponding analyses in the revised paper.
>
> Q2-3: Why not using dilated convolutions instead of or complementary with the mask branch or other design choices from this paper?
> A2-3: First of all, we investigated the usage of dilated convolutions in mask branch before and found that it didn’t make obvious difference. Dilated convolution may be a good choice to obtain spatial attention, as done in BAM [R1]. While, in this paper, we target to obtain non-local mixed attention, including channel and spatial attention simultaneously.
> [R1] Park, Jongchan, et al. "BAM: bottleneck attention module." BMVC 2018.
> Furthermore, we provide more experiments using dilated convolutions in mask branch to demonstrate our above claims. Here we give a brief introduction to the experiments. As dilated convolutions are good at obtaining larger receptive field size, we remove all the non-local blocks in our network. We divide the experiments as 4 cases.
> Case-1: we replace the mask branch with two dilated convolutions and remove our proposed residual attention learning (in Section 3.3 of the main paper) strategy. Namely, we use Eq. (7) for attention learning.
> Case-2: we replace the mask branch with two dilated convolutions and keep our proposed residual attention learning (in Section 3.3 of the main paper) strategy. Namely, we use Eq. (8) for attention learning.
> Case-3: we add two dilated convolutions in the previous mask branch and remove our proposed residual attention learning (in Section 3.3 of the main paper) strategy. Namely, we use Eq. (7) for attention learning.
> Case-4: we add two dilated convolutions in the previous mask branch and keep our proposed residual attention learning (in Section 3.3 of the main paper) strategy. Namely, we use Eq. (8) for attention learning.
> We test the performance on Set5 for color image denoising with noise level=30. To save training time, we set path size as 48, block number as 7. The performance comparisons (in terms of PSNR (dB) within 200 epochs) are as follows:
> Case-1: 31.486 dB; Case-2: 31.508 dB; Case-3: 31.535 dB; Case-4: 31.552; RNAN: 31.602 dB.
> Compare Case-1 and -2, or Case-3 and -4, we can see that our proposed residual attention learning is more suitable for image restoration and contributes to the performance.
> Compare Case-2 and RNAN, we find that mix attention works better than simple spatial attention.
> Compare Case-4 and RNAN, we find that non-local block helps to learn better attention by taking long-range dependencies between pixels than that with dilated convolutions.

---

### Meta-Review · Area_Chair1 · 2018-12-13
**somewhat limited novelty but significant advancement of SOTA**

**Confidence:** 4
**Recommendation:** Accept (Poster)

**Metareview:**

1. Describe the strengths of the paper.  As pointed out by the reviewers and based on your expert opinion.

- strong qualitative and quantitative results
- a good ablative analysis of the proposed method.

2. Describe the weaknesses of the paper. As pointed out by the reviewers and based on your expert opinion. Be sure to indicate which weaknesses are seen as salient for the decision (i.e., potential critical flaws), as opposed to weaknesses that the authors can likely fix in a revision.

- clarity could be improved (and was much improved in the revision).
- somewhat limited novelty.

3. Discuss any major points of contention. As raised by the authors or reviewers in the discussion, and how these might have influenced the decision. If the authors provide a rebuttal to a potential reviewer concern, it’s a good idea to acknowledge this and note whether it influenced the final decision or not. This makes sure that author responses are addressed adequately.

No major points of contention.

4. If consensus was reached, say so. Otherwise, explain what the source of reviewer disagreement was and why the decision on the paper aligns with one set of reviewers or another.

The reviewers reached a consensus that the paper should be accepted.